# Physiological and Molecular Responses of *Vitis vinifera* cv. Tempranillo Affected by Esca Disease

**DOI:** 10.3390/antiox11091720

**Published:** 2022-08-30

**Authors:** José Antonio García, Inmaculada Garrido, Alfonso Ortega, Jerónimo del Moral, José Luis Llerena, Francisco Espinosa

**Affiliations:** 1Centro Tecnológico Nacional Agroalimentario “Extremadura” (CTAEX), Ctra. Villafranco-Balboa 1.2, 06195 Badajoz, Spain; 2Grupo Investigación Fisiología y Biología Celular y Molecular de Plantas (BBB015), Facultad de Ciencias, Campus Avenida de Elvas s/n, Universidad de Extremadura, 06071 Badajoz, Spain; 3Grupo Investigación Calidad y Microbiología de los Alimentos (AGAO17), Instituto Universitario de Investigación de Recursos Agrarios (INURA), 06071 Badajoz, Spain

**Keywords:** chalcone synthase, esca, phenols, phenylalanine ammonia lyase, polyphenol oxidase, systemic infection, superoxide dismutase, FRAP, *Vitis vinifera*

## Abstract

Esca is a multi-fungal disease affecting grapevines. The objective of the study was to evaluate the physiological and molecular response of the grapevine cv. Tempranillo to esca disease, carried out in a vineyard under Mediterranean climatic conditions in western Spain. The photosynthetic pigments in the leaves decreased, with a strong decrease in the photosynthetic efficiency. The proline content increased significantly in the early stages of affected leaves, being possibly involved in the maintenance of lipid peroxidation levels in leaves, which do not increase. The phenol, flavonoid, and phenylpropanoid content decreased in esca-affected leaves, as does the total antioxidant capacity (FRAP), while the polyphenol oxidase (PPO) activity suffers a strong increase with the development of the disease. In affected grapes, the lipid peroxidation and the total phenol content decrease, but not the anthocyanin content. The ascorbate pool decreases with the disease and with time. On the other hand, pool GSH + GSSG is lower in affected leaves, but increases with time. These alterations show a clear change in the redox homeostasis. The expression of genes *phenylalanine ammonia lyase* (*PAL*), *polyphenol oxidase* (*PPO*), *superoxide dismutase* (*SOD*), and *chalcone synthase* (*ChaS1* and *ChaS3*) become considerably higher in response to esca, being even higher when the infection time increases. The alteration of AsA and GSH levels, phenolic compounds, PPO activity, proline content, and FRAP, together with the increase of the *PAL, PPO, SOD,*
*ChaS1,* and *ChaS3* gene expression, are clearly implicated in the esca response in plants. The expression of these genes, similar to the PPO activity, can be used as markers of state in the development of the disease.

## 1. Introduction

Grapevine trunk diseases (GTDs) constitute a major sanitary problem for viticulture worldwide. Both chemical and biological agents, as well as remedial surgery and basic practical viticultural measures, have been proposed and tested for controlling, and especially preventing, the problem [1,2,3,4,5,6]. However, there is currently no effective therapeutic procedure against esca, and this threatens the viability of viticulture, as well as the wine industry deriving from it [6]. In Spain, there are vineyards where its impact can exceed 20% of infected vines, producing severe economic losses. This is the case for vineyards located in *Tierra de Barros* (Extremadura’s emblematic wine-producing region located in western Spain) [7].

Esca is a frequent GTD in Spain, although episodic and prone to appear in mature strains. *Eutypa* dieback, another grapevine disease, appeared in Spain in 1979, with Extremadura being the first autonomous community in which it was identified [8]. Over the last two decades, a general rise of GTD diseases has been confirmed, not only of esca and *Eutypa* dieback, but also others such as *Botryosphaeria* dieback, Petri disease, and black-foot disease [9,10]. These last three seem to affect young vineyards, while esca and *Eutypa* dieback do so with older vineyards [11]. Petri pathogens are associated with esca in mature vineyards as well, and this co-infection is known as the esca complex [11,12,13].

These GTDs are linked to more than a hundred fungi species, many of which are involved in the development of their respective diseases [10,12,14,15,16,17]. Studies have suggested that, in the case of esca, *Phaeomoniella chlamydospora* and *Phaeoacremonium minimum* cause vascular symptoms and would act as primary pathogens, while the basidiomycete *Fomitiporia mediterranea* would cause the decay and death of the wood.

Esca symptoms appear inside the trunk and the main branches (white rot and wood discolouration), the shoots and branches, the leaves (interveinal chlorotic and irregular spots), and the berries (browning and spots) [18]. The responsible pathogens are found in woody tissues of the perennial organs and, to a lesser extent, in annual shoots, but never on the leaves [19,20,21,22]. It is possible that the symptoms observed in leaves and berries are caused by compounds produced by pathogens and/or the affected wood, and transported through the xylem [18,23].

In leaves, the symptoms might be minor, with the development of chlorotic interveinal areas which turn necrotic, or grave, with a sudden withering in a few days. In berries, maturation is delayed, and some dark and purple spots might appear before maturation and then turn necrotic, with this being possible even in plants with asymptomatic leaves [24].

As a consequence of the disease, the carbon reserves decline, which can cause a decrease in the plant’s development and vigour in the following year. The metabolism of lipids and amino acids is affected as well. Disturbance in these primary metabolisms is often linked to the induction of defence responses [24].

Compounds produced by fungi could activate the secondary metabolism involving the synthesis of anthocyanins and the key enzymatic reactions (NADPH oxidase and phenylalanine ammonia lyase). Toxic polypeptides from fungi can modify the metabolism of plant cells through different pathways [25]. Fungi can produce oxidative enzymes such as laccases (a type of PPO together with creolase and catechol oxidase [26]) which can oxidize phenolic and polyphenolic compounds [27]. Goufo et al. [5] highlight how the different responses of the phenolic compounds to the esca complex can be linked to the types and complexities of the symptomatic and asymptomatic materials that various workers have studied, such as the different behaviours of the varieties, weather conditions, and cultivation regimes [5,28,29,30].

The presence of symptoms and the subsequent performance losses may vary from one season to the next, regardless of the rate of woody tissue decay. These symptoms could occur in a vine during one growing season but not systematically in the next, and always taking into account physiological, cultural, and environmental factors. The spread of wood diseases seems to thrive on global warming conditions, especially due to the more powerful heatwaves and droughts [31]. Moreover, the appearance of symptoms seems to be linked to the nutritional state of the affected vineyard, in particular with the amount of Ca, while the use of fertilizers with Ca and Mg reduces the expression of symptoms [32].

In this work, we analysed the responses of *Vitis vinifera* cv. Tempranillo leaves and grapes to the esca disease. The contents of photosynthetic pigments, phenol, proline, ascorbate, and glutathione were determined. Lipid peroxidation, photosynthetic efficiency, total antioxidant capacity, and polyphenol oxidase activity were also evaluated. In order to evaluate the plant’s possible antioxidant response against fungi which cause the disease at the physiological and molecular level, the transcription levels of genes related to phenolic metabolism and defence reactions were also assessed.

## 2. Materials and Methods

### 2.1. Plant Material and Experimental Design

The experiment was carried out in a 16-year-old vineyard, located at Finca la Orden (Regional Government of Extremadura, Badajoz, Spain) (38 °N, 6 °W, elevation 198 m), under Mediterranean climatic conditions. The vineyard was planted in 2001 using *Vitis vinifera*, L., cv. “Tempranillo”, grafted on Richter-110 rootstock at a spacing of 2.5 m by 1.2 m (3333 grapevines·ha^−1^). The soil at the site had a loam to sandy-loam texture. The volumetric water content was 20.4% at field capacity and 11.4% at permanent wilting point. Table 1 shows the climatic conditions in the experimental vineyard in 2017.

Ten leaves from five healthy grapevines (showing no esca symptoms in the last 3 years) and five affected by esca (with visible symptoms, in tiger-like stage) were harvested on 12 June and 20 August 2017. Two bunches of grapes from five healthy and five esca-diseased grapevines were harvested in the October 2017 crop. All leaves or bunches from the same treatment were mixed. All samples, once collected, were immediately frozen with liquid nitrogen and kept at −80 °C until the start of the analysis. From each of the treatments (healthy and esca-diseased) and harvest dates, 10 samples were made to carry out, in triplicate, each of the biochemical or molecular determinations.

Healthy and esca disease leaves were collected from the same grapevines (10 samples of each experimental condition) on two different dates: 12 June and 20 August 2017. The grapes belonged to the October 2017 crop, also from healthy and esca disease vineyards. All samples, once collected, were immediately frozen with liquid nitrogen and kept at −80 °C until the start of the analysis.

### 2.2. Photosynthetic Pigment Contents and Photosynthetic Efficiency 

Leaf discs from fresh leaves were taken and incubated in methanol (12.5 mg mL^−1^) for 24 h in darkness at room temperature. The chlorophyll a, chlorophyll b, and carotenoid contents were determined in a spectrophotometer (Shimadzu UV 1603, Kioto, Japan) by measuring A_666_, A_653_, and A_470_, expressed as µg g^−1^ FW. The total chlorophyll and carotenoid contents were calculated following Wellburn [33].

The maximum photosynthetic efficiency (F_V_/F_M_) was determined on fresh leaves of intact plants, before being collected, using a “ChlorophyllFluorometer OS-30p” device (Opti-Sciences, Hudson, NH, USA). Prior to the excitation, the leaves being sampled were kept in darkness for 10 min, then illuminated so as to measure the fluorescence emitted and calculate the F_V_/F_M_ ratio [34].

### 2.3. Determination of Lipid Peroxidation

The peroxidation of membrane lipids was determined spectrophotometrically from the formation of MDA (malondialdehyde) from TBA (2-thiobarbituric acid). To this end, 0.1 g of leaves were homogenized in 1 mL 0.25% TBA and 10% TCA (trichloroacetic acid), incubated at 95 °C for 30 min, filtered through muslin cloth, and centrifuged at 8800× *g* for 10 min [35]. The grapes, 0.2 g, were homogenized in 1 mL 0.1% TCA, then centrifuged (10,000× *g* 10 min), and the supernatant was incubated at 20% TCA and 0.5% TBA 95 °C for 30 min, and then centrifuged (10,000× *g* 15 min). The amount of MDA was determined from A_532_–A_600_ with the extinction coefficient ε = 155 mM^−1^ cm^−1^, with the result expressed as μmol MDA g^−1^ FW [36].

### 2.4. Determination of Proline Content

The proline content was determined in accordance with the method of Bates et al. [37]. Briefly, 0.5 g of leaves or grapes were homogenized in 2.5 mL of 3% sulfosalicylic acid, filtered through muslin cloth, centrifuged at 10,000× *g* for 10 min, and 500 μL of the supernatant was added to a mixture of the same volumes of glacial acetic acid and ninhydrin. The resulting mixture was incubated at 100 °C for 1 h, then placed into ice to stop the reaction. To each reaction tube, 1.5 mL of toluene was added, followed by vortexing for 20 s. After 5 min left at rest, the absorbance at 520 nm was measured in a spectrophotometer. The concentration of proline was calculated from a standard curve, expressing the result as μg proline g^−1^ FW.

### 2.5. Phenolics Content and PPO Activity

Phenols, flavonoids, and phenylpropanoid glycosides were extracted from 0.2 g fresh leaves and grapes by homogenization in 2.5 mL of methanol, chloroform, and 1% NaCl (1:1:0.5), filtering through muslin cloth, and centrifuging at 3200 g for 10 min. Total phenols were determined spectrophotometrically at A_765_ with the Folin–Ciocalteu reagent [38], expressing the result as μg caffeic acid g^−1^ FW. Total flavonoid content was measured at A_415_ [39], expressing the result as μg of rutin g^−1^ FW. Phenylpropanoid glycosides were determined at A_525_ [40], expressing the result as μg verbascoside g^−1^ FW. In all cases and in order to quantify the content in the different compounds, a corresponding standard curve was made.

The content of anthocyanins was quantified according to Giusti and Wrolstad [41]. Briefly, 2.5 g of grapes were homogenized in 2.5 mL in a solution of ethanol:HCl 0.1 M (85:15% *v*:*v*), and it was centrifuged at 4000× *g* for 10 min; the supernatant was diluted in different buffer solutions (KCl 0.025 M, pH 1.0 and sodium acetate 0.4 M, pH 4.5), and incubated for 30 min at room temperature. Absorbance was measured at 520–700 nm. The content of anthocyanins was expressed as mg of malvidin 3-glucoside g^−1^ FW from the corresponding standard curve [42].

For the PPO activity, leaves or grapes samples were homogenized at 4 °C (0.5 or 0.15 g mL^−1^, respectively) in 100 mM phosphate buffer (pH 6.5), 1% PVPP. The homogenate was filtered through muslin cloth and centrifuged at 19,000× *g* for 30 min at 4 °C. The filtered supernatant was immediately used for assay. The pellet was discarded, and the supernatant filtered for the assays and protein content determination [43]. PPO activity was determined by measuring A_420_ at 30 °C in a medium containing the extract, 100 mM phosphate buffer, and 0.1 M catechol [44]. A unit of PPO activity was defined as the amount of enzyme required to cause a ∆A_420_ of 0.001 units min^−1^.

### 2.6. Determination of Antioxidant Capacity

Ferric reducing antioxidant power (FRAP) determination was performed at A_593_, as described in Rios et al. [45]. Calibration was done against a standard curve using freshly prepared ammonium ferrous sulfate [46], and the concentration was expressed as μg of ferrous sulfate g^−1^ FW.

### 2.7. Ascorbate and Glutathione Contents

To determine the ascorbate (AsA), dehydroascorbate (DHA), reduced gluthathione (GSH), and oxydized gluthatione (GSSG) contents, leaves or grapes (1 g mL^−1^) were homogenized at 4 °C in 5% metaphosphoric acid in a porcelain mortar. The homogenate was filtered through muslin cloth and centrifuged at 16,000× *g* for 20 min at 4 °C. The total ascorbate and glutathione assays were done in accordance with De Pinto et al. [47]. The total ascorbate pool was determined in a reaction medium containing the extract, 150 mM phosphate buffer (pH 7.4), and 5 mM EDTA, which was incubated for 15 min in darkness. The result was then complemented with 0.5% NEM (N-ethylmaleimide), 10% TCA, 44% orthophosphoric acid, 4% dipyridyl, and 110 mM FeCl_3_, followed by incubation at 40 °C for 40 min in darkness. The reaction was halted with ice, and the A_525_ was spectrophotometrically measured. To determine the amount of AsA, 10 mM DTT (DL-dithiothreitol) was added to the reaction medium before incubation in darkness, while 100 μL of water was added to determine the ascorbate pool. The concentration of DHA was estimated from the difference between the total ascorbate pool (AsA + DHA) and AsA. 

The total glutathione pool was determined by adding 0.4 μL of extract to 0.6 μL of 0.5 mM phosphate buffer (pH 7.5). The reaction medium containing the extract, 0.3 mM NADPH, 150 mM phosphate buffer (pH 7.4), 5 mM EDTA, and 0.6 mM DTNB (5,5′-dithiobis-(2-nitrobenzoic acid)) was stirred for 4 min, then 2 U mL^−1^ GR was added and the A412 was measured. To determine the GSSG content, the mixture was incubated for 1 h in darkness with 2- vinylpyridine (20 μL) to eliminate GSH, and, to determine the glutathione pool, 20 μL of water was added. The amount of GSH was obtained by the difference between the total pool (GSH + GSSG) and the amount of GSSG.

### 2.8. RNA Extraction and Synthesis of cDNA

The plant material (healthy and esca-diseased) collected was in the same development state (adult leaves and grapes) and was contained in aseptic plastic bags; after its collection, it was immediately frozen with liquid nitrogen and kept at −80 °C until the start of the extraction process. The homogenization was carried out in mortars until each sample was reduced to dust, and this homogenized material was transferred to Eppendorf^®^ tubes until reaching, approximately, 100 mg of fresh weight. All these actions took place using sterilized, clean, and RNAse-free material.

Total RNA purification was carried out with the “Spectrum Plant Total RNA” kit from Sigma-Aldrich^®^ (St. Louis, MO, USA) and using the DNAsa “RNase-Free DNase Set” (Cat No 79254) from QIAGEN^®^ (Hilden, Germany) The concentration and pureness of RNA dissolved in elution buffer given in the kit (free from RNAse and DNAse) was quantified with an BioSpectrometer (eppendorf^®^, Hamburg, Germany) Only those in which the ratio 260/280 nm presented a value between 1.8 and 2.0, as it was obtained in each extraction, were considered quality samples. The integrity of the extracted RNA was assessed using agarose gel electrophoresis at 1.5% of Duchefa^®^ with a buffer solution of TAE (Tris-Acetato-EDTA) 1× of Fisher reagent^®^ and with ethidium bromide from Sigma-Aldrich^®^ as intercalating agent (0.075%), adding 2.5 µL RNA after 2 µL of Thermo-Fisher^®(^ (Waltham, MA, USA) buffer solution. The gel was visualized in a transilluminator Geneflash (Syngene^®^, Cambridge, UK), with two well-defined bands observed, coming from the transfer RNA (tRNA) 28 s (upper band) and 18 s (lower band). The total RNA samples were kept at −80 °C.

Each RNA was transformed into its corresponding cDNA, whose synthesis was carried out using the High Capacity cDNA Reverse Transcription kit from Applied Biosystems^®^ (Waltham, MA, USA). The cDNA was originated from 1–2 μg of each RNA sample, with a concentration of, approximately, 100 ng μL^−1^. Having done the reverse transcription containing the primers (random primers) which were included in the kit, a T100^TM^ Thermal Cycler from BIO RAD^®^ (Hercules, CA, USA) was programmed with the first phase of 10 min at 25 °C to hybridize the primers, followed by another phase at 37 °C for 120 min to let reverse transcriptase act and, finally, the inactivation phase of the reaction, maintaining a temperature of 85 °C for 5 min. The cDNA obtained was distributed in aliquots and stored at −20 °C.

### 2.9. Identification and Amplification of the Genes Studied

With an aliquot of each cDNA sample, high-fidelity polymerase chain reactions (PCR) were performed using the Taq Polymerase HiFi enzyme from Applied Biosystems^®^ for the identification and evaluation of the expression level of *ChaS1*, *ChaS3*, *PAL*, *PPO,* and *SOD*. The primers are shown in Table 2. They were designed in the IDT portals (Integrated DNA Technologies^®^, Newark, NJ, USA) and Primer3Plus from the information drawn during the coding sequences, which were offered by GRAMENE and NCBI databases; or using primers published in previous works (all this information can be found in Table 2). 

The different PCR were programmed in the thermal cycler aforementioned under the following conditions: 62 °C annealing temperature, with a period of time of 30 s, given that the size of each amplicon was not larger than 100 base pairs (bp). The products of each PCR were mixed with 2 µL Thermo-Fisher^®^ loading buffer and were loaded with an agarose gel at 1% with TAE (Tris-Acetate-EDTA) 1× buffer solution and with ethidium bromide as an intercalating agent (at 0.075%). After electrophoresis, the gels were revealed under UV light in a Geneflash (Syngene^®^, Cambridge, UK) transilluminator. The gels were photographed using a digital camera (768 × 582 pixels and 8 bits), and the images were printed with a Video Graphic Printer from Sony^®^ and saved as TIFF files. As an indicator of the DNA fragment size, we used a *Ladder 1 Kb plus* from Thermo-Fisher^®^. After verifying that the size of the bands in each sample in the gel was accurate, we started the process of quantitative polymerase chain reaction (qPCR).

For the different qPCR, SYBR green qPCR (Thermo-Fisher^®^ (Waltham, MA, USA) was used, and the detection and quantification of the amplification was carried out using a PCR LightCycler^®^ 480 II (Roche diagnostics, Basel, Switzerland). Each gene studied was paired with two different reference genes (*ACT2* [At3g18780] and *UBQ10* [At4g05320]). The values of the expression were calculated with the efficiency method of LightCycler^®^ 480 software, version 1.5 (Roche Diagnostics, Basel, Switzerland). The qPCR was carried out using housekeeping *VATP16* (V–type proton ATPase) [48] and *Actin2* [49,50], to evaluate both the expression level and the success of the extractions.

Finally, we analyzed the coding and peptide sequences of the enzymes ChaS1, ChaS2, and ChaS3. We obtained these sequences through the GRAMENE database (www.gramene.org, accessed on 15 May 2021). This analysis consisted of a global alignment of the nucleotide coding sequence and the amino acid sequence using Geneious version 7.1 software, Geneious R8 plataform (San Diego, CA, USA). For these alignments, we used the Blosum90 matrix (BLOcks of amino acid Substitution Matrix; with a maximum identity of 90%) with “free end gaps”, and the genetic distance was calculated using the Jukes–Cantor model [51]. Phylogenetic analysis was performed using the Jukes–Cantor model and neighbor-joining as a statistical model [52].

### 2.10. Statistical Analyses 

The physiological and biochemical results are the means ± SD, and a Mann–Whitney U test was performed for statistical significance (*p* < 0.05). A Student’s t-test was performed in order to determine whether there were significant differences (*p* < 0.05) in the gene expression between healthy and esca-diseased grapevine plants. We performed the statistical analyses with Microsoft Excel (Redmond, Washington, USA) and IBM^®^ SPSS^®^ vn 21.0 Statistics software package, Chicago, IL, USA).

## 3. Results and Discussion

### 3.1. Pigments and Photosynthetic Efficiency

Both chlorophyll a and b contents decreased drastically in leaves with visible symptoms of esca (esca-diseased) (Table 3). In the leaves sampled in June, it was observed that those with visible symptoms of esca had chlorophyll a and b values representing 51% and 47%, respectively, of the values obtained in healthy leaves. The total chlorophyll content decreased by 50% compared to the value observed in healthy leaves. With regard to the evolution in chlorophyll a and chlorophyll b content, in the August-collected leaves, the values in the healthy leaves decreased, while in symptomatic leaves they stayed similar to the June-collected values, although always being below those of the healthy leaves. While in the healthy leaves the chlorophyll a/b ratio stayed constant between June and August, in the symptomatic leaves this ratio was higher on both dates, and slightly greater in June than in August (2.16 vs. 2.05). A similar tendency was observed for the carotenoid content, since healthy leaves presented a greater value for this parameter in June than the symptomatic leaves (whose content was 62% of that of the asymptomatic leaves). It was interesting that in the August sampling of both types of leaves, the carotenoid content was lower and similar for the two types. The carotenoid/chlorophyll ratio was significantly greater in esca-diseased leaves than in healthy ones.

In the healthy August leaves, there was a decrease in chlorophyll, a fact that reflects the evolution of these pigments with the phenological state. When the berries are in their veraison period, a decrease in chlorophyll content begins [53]. This decrease also occurs with increasing temperature [54], as was the case with our samples. In plants with esca-disease, the leaves may show decay due to the response to the pathogen. We could also observe chlorosis and necrosis in these leaves. The increase in the chl a/b ratio in leaves of plants with esca disease could be due to biotic stress. However, the chl a/b ratio in leaves of plants with esca disease in August was similar to the values measured in leaves of healthy plants. Our hypothesis to explain this fact is that there is an overlap with drought stress. On the other hand, in both June and August, the carotenoid/chlorophyll ratio was greater in leaves from plants with esca-disease than in leaves from healthy plants. The increase in carotenoids is a response to stress, which allowed the June photosynthetic efficiency to remain unchanged, although in August both carotenoids and chlorophylls had decreased, but less than in the leaves of healthy plants.

These decreases in chlorophyll and carotenoid contents are in agreement with those described by Martin et al. [30] in this same grape variety at three different locations, and by Bertamini et al. [55], Petit et al. [56], and Rusjan et al. [57] in cv. Chardonnay, and Valtaud et al. [58] in cv. Ugni Blanc. Santos et al. [59] inoculated in vitro two fungi involved in the development of esca using different cultivars, and described a decrease in the chlorophyll content, probably because the chloroplasts are one of the organelles where catabolism first starts at the beginning of senescence [60]. On the other hand, Magnin-Robert et al. [61] observed changes in neither the chlorophyll nor the carotenoid content, nor any alteration in the a/b chlorophyll ratio. In our case, the a/b chlorophyll ratio decreased slightly, while the total carotenoid/chlorophyll ratio increased in esca-diseased leaves with respect to the healthy ones. However, Bertamini et al. [55] describe a clear decrease in a/b chlorophyll ratio in esca-diseased leaves compared to healthy ones, although they do report an increase of the total carotenoid/chlorophyll ratio. Rusjan et al. [57] and Valtaud et al. [58] observed an unaltered a/b chlorophyll ratio. Regarding the photosynthetic efficiency, there was a strong decrease in symptomatic leaves, being greater in the August-collected leaves, which are in a much more advanced state of the disease. This result matches that described by Bertamini et al. [55], Magnin-Robert et al. [61], and Letousey et al. [62] in the Chardonnay variety. The a/b chlorophyll ratio is inversely proportional to the degree of appression of the thylakoid membranes in the chloroplast. With greater appression, the light-harvesting complex II is more closely linked, thus increasing the capture of light, and therefore the energy transmission efficiency. Increased appression of the thylakoids improves electron transport from photosystem II to the cytochrome b_6_f complex [63]. Our data are indicative of how affected leaves are less effective in harvesting light and the transmission of electrons, resulting in lower photosynthetic efficiency.

### 3.2. Lipid Peroxidation and Proline Content

There are no visible changes in lipid peroxidation in the leaves, with very similar values in both June and August in both the healthy and esca-diseased cases (Table 4). However, in the berries there is a decrease in lipid peroxidation in the esca-diseased leaves when compared to the healthy ones (Table 5). Santos et al. [59], in plants and calluses cultivated in vitro and inoculated with fungi involved in the development of the disease, observed an increase in lipid peroxidation that would affect the integrity of the membranes, a symptom of senescence. In the present study, there were no changes in the levels of peroxidation, probably due to the SOD activity, which showed a strongly increased expression. The action of the enzymatic antioxidant system (SOD, POX, and enzymes of the ascorbate/glutathione cycle) may be responsible. The decrease in ROS levels and activity, together with PPO to produce lignin and lignans, could be involved. Regarding the proline content, an amino acid involved in different defence mechanisms of plants (mostly against water stress), it was observed that esca-diseased leaves from June showed values far greater than the healthy ones (Table 4).

Ozden et al. [64], in the cv. Oküzgü exposed to oxidative stress by applying H_2_O_2_, observed how proline accumulation can limit the diffusion of H_2_O_2_, and, with it, lower the lipid peroxidation in the membrane. Nevertheless, in our August leaves, the proline content was very similar in healthy and esca-diseased leaves, and far greater than the values obtained in the June leaves, possibly as a consequence of a certain degree of water stress produced on this date. In the berries, the proline content was similar in both the healthy and the esca-diseased cases (Table 5). Proline may also be involved in the response to pathogens [65,66]. In June leaf samples, the amount of proline was greater in leaves of esca-diseased plants than in those of healthy plants. This increase in the amount of proline may have been due to the response to pathogen attack. However, in August, the proline value was very similar in both types of leaves (from esca-diseased and from healthy plants). This similarity was probably due to the response to water stress and high temperature, in addition to the effect of esca. The proline content seems to be more related to the plant’s response to water stress than to the esca itself. Moreover, Yan et al. [67] observed how proline can act to protect the chlorophylls and photosystem II, which might explain the lower values we obtained in the June-harvested leaves regarding the effect of esca on photosynthetic efficiency, with a decrease of 17% compared to the decrease of 39% in the photosynthetic efficiency in the August-harvested leaves. Proline could act by protecting the cell membrane from the lipid peroxidation processes and favouring the elimination of ROS, both directly and through the stimulation of the enzymatic antioxidant systems involved in this process [64,68,69]. In the cv. Tempranillo leaves of our research, an increase in proline was observed in the June leaves, which could be related to the lack of alteration in the lipid peroxidation levels. In the August leaves, the effect of esca could act in combination with the water stress, with high levels of proline in both healthy leaves and esca-affected ones, although in neither case did they show high lipid peroxidation levels.

### 3.3. Total Phenols, Flavonoids, Phenylpropanoid Glycosides (PPGs), Anthocyanins, Antioxidant Capacity (FRAP), and Polyphenol Oxidase (PPO) Activity

Total content of phenols, flavonoids, and PPGs (Figure 1) in the June-harvested leaves did not significantly differ between healthy and esca-diseased leaves. On the contrary, in the August-collected leaves, the ones with esca presented lower levels of all phenolic compounds. Specifically, the phenols were lower by 11%, flavonoids by 15%, and PPGs by 13%. These results match those described by Martin et al. [30] of decreases in the phenol and flavonoid contents in cv. Tempranillo as a consequence of the infection. Moreover, they also described changes in the phenolic profile, although this could have been a consequence of the different environmental and physiological conditions, as well as or instead of the esca disease. On the contrary, Lima et al. [70] describe an increase of the phenol production in esca-affected leaves. Moreover, Goufo et al. [5], in cv. Malvasia, detected an increase in total flavonoids and phenols only in esca-diseased leaves in the early state of the disease, compared with healthy leaves; although, as the severity of the chlorosis and necrosis increased, these compounds declined below the control values of healthy leaves. There is a wide variability in the accumulation of phenolic compounds in response to esca. This variability indicates that there are transient and dynamic changes at the metabolic level as the disease progresses. All this can lead to alterations in the content of phenolics, allowing some phenolic compounds to accumulate instead of others [5,71].

In berries from esca-diseased plants, the total content of phenolic compounds is less than that observed in the healthy cases, with values that represent decreases of 42%, 23%, and 35% for phenols, flavonoids, and PPGs, respectively (Table 5). Lorrain et al. [29] also described a lower quantity of phenols in the skin of the esca-diseased grapes. These phenolic compounds could be involved in the defence mechanisms and be degraded, which could explain this decrease. Moreover, in oxidative stress conditions, there could be some changes in the phenolic metabolism, causing the synthesis of stilbenoids, lignins, lignans, and neolignans [71]. Nevertheless, Calzarano et al. [28] detected a polyphenol increase in grapes, must, and wine coming from symptomatic grapevines.

Anthocyanins, the main flavonoid group of grapes, are associated with the organoleptic properties of red wine, such as colour and astringency. The anthocyanin content in berries (Table 5) is practically the same in healthy plants and esca-diseased ones, with the non-significant small decrease observed in the ones affected by the disease possibly due to delayed ripening because of the disease. Lorrain et al. [29] indicated that esca causes a decrease in the anthocyanin content in the skin of the grape, which could be due to the disease affecting the physiology of the carbohydrates, the transport of water through the xylem, and photosynthesis, with its obvious adverse consequence for carbon metabolism [56,72]. The synthesis pathway of flavonoids responsible for the synthesis of tannins and anthocyanins could be affected, leading to a lesser phenolic compound content.

Regarding the total antioxidant capacity (Figure 1), closely linked to the content of phenolic compounds, healthy leaves showed a similar antioxidant capacity in those collected in June and in August. On the other hand, the esca-diseased leaves exhibited a decrease as the infection process advanced. Thus, in June, the affected leaves had an antioxidant capacity representing a 12% reduction with respect to the healthy ones, while in August, this decrease was 45%. In healthy berries, high values were observed, far greater than those of the esca-diseased berries (Table 5), with the latter showing a total antioxidant activity which was 49% of the healthy case. Atak et al. [73] showed in *Vitis* spp. leaves infected with *Uncinula necator* and *Plasmopara viticola* fungi that there is an increase in phenol content and antioxidant FRAP capacity. Our results showed a clear relationship between phenolic compound content and total antioxidant capacity (FRAP) [74]. Thus, the greater content of phenolic compounds in healthy leaves and berries was related to a greater antioxidant capacity than in the esca-diseased cases.

The PPO activity during the initial stages of esca disease in the June-collected leaves was at similar levels to the healthy cases (Figure 2). However, as the diseases advanced, in the autumn-collected leaves it was observed that, while in healthy leaves this activity maintained similar values, in affected leaves there was a strong increase of PPO activity, which could be up to 11× greater than the value observed in the healthy cases and in the esca-diseased June leaves from green vine stems. In berries (Table 5) collected in August, there was a similar increase. Healthy berries presented a lower PPO activity than those from the esca-diseased vineyard, with a 7-fold increase in this activity. This PPO activity increase was also observed by Rusjan et al. [57]. Spagnolo et al. [75] detected more PPO expression in green vine stems from esca-diseased grapevines. PPO activity would be involved in the resistance of the plant through the production of metabolites that are toxic for the pathogens, such as phytoalexins, phenols, and lignin. Pasquier et al. [76] also found more PPO protein in esca-diseased berries than in healthy ones.

### 3.4. Ascorbate and Glutathione Content

Ascorbate and glutathione are two important antioxidant molecules involved in the redox homeostasis. With regard to AsA (Figure 3), the results show a greater concentration in June-collected leaves, with the lower content in August being 54% of that observed in June. Although the effect of esca was similar on both dates, there was a decrease in the AsA content by 20% and 35% for June and August, respectively. In addition, DHA content declines with time. Esca induces decreases in DHA in leaves of 18% and 40% in June and August, respectively. As a consequence, the total ascorbate content is greater in healthy June-collected leaves than in the August ones, with the healthy values always being greater than the esca-diseased ones. The AsA/DHA ratio (redox state) was similar in healthy and esca-diseased leaves in June (0.87 and 0.86, respectively), and showed a slight but non-significant difference in August (1.37 and 1.44, respectively). The ratio was greater in both the healthy and the esca-diseased cases in the August-collected leaves than in those from June. The decrease in AsA content was also observed by Bortolami et al. [77], although in that work the DHA content increased, while in our esca-diseased leaves there was a decrease. The AsA and DHA behaviours are dissimilar. Thus, Kuźniak and Skolowska [78] describe how, in tomatoes infected by Botrytis, while the AsA content stays stable at similar values in healthy and affected leaves, DHA increases in response to the infection. On the contrary, Sgherri et al. [79], in cv. Trebbiano, describe a completely different behaviour in response to the viral infection, observing increases in AsA and decreases in DHA, but an unchanged redox state (AsA/DHA ratio). Other researchers have found dissimilar behaviours; Bruno et al. [80] observed a lowered AsA/DHA ratio, while Sgherri et al. [79] described a rise. This different behaviour of AsA in leaves might reflect the different varieties used, the different stages of infection, or even the interaction of phenols and flavonoids with the AsA to act as an H_2_O_2_ elimination system [81]. The behaviour of phenols and flavonoids in response to infection is also a matter of controversy, and differs according to the variety and to environmental conditions [77].

With respect to glutathione, the GSH + GSSG pool decreased in esca-diseased leaves compared to healthy leaves, in both June and August (Figure 4). The total content fell to values representing 72% and 63% of the control values in June and August, respectively. In both healthy and esca-diseased leaves, the glutathione pool increased with time, being greater in August than in June. The GSH content showed a similar evolution. Esca-diseased leaves had lower values than control leaves, 60% and 63%, for both harvesting dates. However, the evolution of GSSG content was somewhat different. In esca-diseased leaves, the decrease was significantly lower than that observed for GSH, with values representing 67% and 68% of the GSSG content of healthy leaves. The value of the GSH/GSSG ratio remained constant, although always below the values of healthy leaves. (2.93 and 2.73, and 2.98 and 2.76, for healthy and esca-diseased leaves, from June and August, respectively). Similar to our results, Valtaud et al. [82] described a decrease in the total glutathione pool, with alteration of the redox state in grapevine leaves as tinder disease developed. On the contrary, Sgherri et al. [79], in grapevine leaves infected by fanleaf virus, observed an increase in total glutathione content in diseased leaves, mainly due to an increase in GSH content, with time-dependent fluctuations of the redox state. In our case, the decrease in the glutathione pool was not compensated by an increase in the ascorbate pool and phenol content, as they also decreased. This behaviour shows a clear effect on the total antioxidant capacity of the diseased leaves, which decreased. There was a clear alteration of redox homeostasis in the esca-affected leaves.

In berries (Table 6), the AsA content was lower by 66% and DHA by 28% in esca-diseased berries compared to the healthy ones. These declines led to a sharp decline (by 55%) in the AsA + DHA pool. There was an imbalance in this ratio, with AsA/DHA being much greater in healthy berries than in affected ones, 2.63 vs. 1.22, indicating a greater antioxidant potential due to this compound in healthy berries. However, the GSH + GSSG pool increased relative to the values observed in healthy berries, with an increase of 16% (Table 6). GSH content was unaffected by esca, but GSSG increased by 38%. These alterations caused the GSH/GSSG redox state to decrease in esca-diseased berries.

### 3.5. Gene Expression

The results related to the expression levels of our target genes were normalized with *VATP16* (V-type proton ATPase), which has been indicated to present a stable expression in foliar vine samples, in both healthy individuals and ones under stress, mainly biotic stress [48,50,83,84]. A single gene (*VATP16*) was used to normalize the results instead of a combination of several, since previous experiments, such as that of Gamm et al. [48], have found it to be more effective and it is a more representative gene given the characteristics and determinants of the present study.

According to the results, all the grapevines which were subjected to stress induced by the development of the esca disease presented levels of expression significantly greater for our target genes *ChaS1*, *ChaS3*, *PAL*, *PPO*, and *SOD* than the control grapevines which were unaffected by esca (Figure 4). The increase in the expression levels depended strongly on the infection time, with the August samples giving greater values than those of June. The *ChaS1* expression was that which grew the most with esca infection time. One must consider that both ChaS’s (involved in the biosynthesis of flavonoids with phenylpropanoids via catalysis of chalcone synthase (ChaS) and the chalcone isomerase (CHI) [85,86,87]) and PAL (which degrades phenylalanine to ammonia and cinnamic acid, and synthesizes salicylic acid) [88] participate in the biosynthesis of phenolic compounds. Nevertheless, a clear difference can be observed in their expression. These different levels of expression could be because one biosynthesis pathway gets more activated than another, or because there are stricter regulating mechanisms in PAL. On the other hand, the biochemical results show that, in esca-diseased plants, the total amount of phenolic compounds (flavonoids, phenylpropanoids, and total phenols) decreases in the August samples when the expression of PPO codifying genes is higher, which could mean that the slight decrease is due to an increase in the said expression, and therefore its activity against phenolic compounds. We must consider that PPO is associated with defence processes in plants, particularly under biotic stress conditions [89,90]. Its function is the degradation of phenolic compounds [91]. PPO seems to be also involved in the production of ROS [92], and could act together with peroxidase (POD) to produce lignins and lignans, which would explain why the lower the total content of phenolic compounds, the greater the *PPO* expression. However, affected leaves show a notable increase in *SOD* expression, with values greater than the controls in both June and August. The *SOD* expression rises with infection time. This result proves the participation of SOD in the defence mechanisms against oxidative stress induced by esca, and is linked to the increase in SOD activity described in grapevines as a response to stress [64,79]. A greater expression of *SOD* can act to ease the harmful effects of O_2_^−^, producing H_2_O_2_, which could also be used in lignification processes. An interaction among SOD, PPO, and POD could be established. This increase contrasts with the results of Magnin-Robert et al. [61] in cv. Chardonnay, where they observed a decrease in the expression levels of *SOD*. Moreover, the increase of PAL activity could increase the stilbenes as a defence mechanism against infection, with their biosynthesis stimulated by ROS produced during the oxidative stress process [71]. Magnin-Robert et al. [61] and Lambert et al. [93] describe a strong increase in *stilbene synthase* (*STS*) expression in response to esca, key to the biosynthesis of these compounds. An increase in the synthesis of stilbenes from 4-coumaroyl CoA could explain why the increase in *CHI* expression does not translate into a clear increase in flavonoids, which can be oxidized by the PPO. Similar increases in *PAL* expression have also been described by Magnin-Robert et al. [61], Letousey et al. [62], and Lambert et al. [93].

These kinds of results agree with those obtained in the analysis of berries harvested in October from healthy and esca-diseased vine plants (Figure 5). Analysing the different samples, it is clear that both the activity and the expression of *PPO* is elevated in berries from grapevines subjected to biotic stress (esca-diseased) compared to berries from healthy grapevines. We determined a 10-fold increase in PPO activity in esca-diseased berries compared to berries from healthy grapevines, and a 6-fold increase in *PPO* expression levels in berries from esca-diseased vine plants. These results agree with other work that has reported the involvement of this enzyme in responses to biotic stress [89,90], and we have been able to confirm that this type of behaviour would happen both at the foliar level and at the fruit level. There is also an increase in the gene expression of *ChaS1*, *ChaS3*, and *PAL*, but not of *SOD*. Phenolic compounds were quantified, and found also to be lower in berries from diseased vine plants. This result again agrees with what had been determined in the foliar organ, and could be due to the greater activity of PPO and the expression of the genes that encode this enzyme, with there being fewer phenolic compounds because it degrades them [91]. Due to the difference in expression between the two chalcone synthases (*ChaS1* and *ChaS3*), we analysed their coding sequences, as well as that of chalcone synthase 2 (*ChaS2*). *ChaS2* was cited for the first time in the work of Goto-Yamamoto et al. [94], and is currently considered a variant of *ChaS1*. However, Gaiotti et al. [95] recently determined that the expression of *ChaS2* remains invariable under stress situations, while *ChaS1* and *ChaS3* present high levels of expression. With this phylogenetic analysis, we wished to see whether there are non-conserved domains between the different ChaS isoforms that explain their unequal response to stress.

*ChaS1* and *ChaS3* show different levels of expression (Figure 4 and Figure 5). Due to these differences in expression, we analysed the coding sequences of the two ChaS isoforms (1 and 3), together with the other isoform described, ChaS2 [95] (Appendix A). *ChaS1* and *ChaS2* are found on the same chromosome (14) and in close-by locations: *ChaS1* at 14: 24 686 995–24 688 542 and *ChaS2* at 14: 24 673 459–24 675 059. The reason the two genes are both on the same chromosome and nearby may be due to a duplication of that region. The result of the alignment of the nucleotide coding sequence, as well as the amino acid sequence, shows a very high affinity, including the coenzyme A binding domains and the catalytic triad [96]. The phylogenetic tree with the amino acid sequence shows a high proximity between *ChaS1* and *ChaS2* and, somewhat further away, with *ChaS3* (Appendix A). A deeper functional characterization of *ChaS2* should be carried out to determine whether there is any key domain that they do not share with the other two isoforms, and thus explain how, in works such as those by Gaiotti et al. [94], the expression levels of *ChaS3* and *ChaS1* are reported to be significantly higher under stress situations than the expression of *ChaS2*, whose expression level remained unchanged under these adverse circumstances. On the other hand, *ChaS3* is located on chromosome 5 (13,597,428–13,599,519). After performing the alignment of the coding sequence and the peptide sequence, we observed that the domains corresponding to the binding to coenzyme A and the catalytic triad [96] coincide with the other isoforms. Therefore, the unequal expression of the two genes cannot be explained by there being truncated functional regions. It needs to be studied whether the non-coinciding regions influence this differential expression of the two genes.

## 4. Conclusions

The foliar-level alterations which the disease induces in the Tempranillo variety are produced physiologically and molecularly. Our results show a greater phenolic compound content and antioxidant capacity in healthy plants than in esca-diseased plants. However, contrarily, the proline content and PPO activity increase. These results could be linked to the molecular expression results of *ChaS*, *PAL*, *PPO*, and *SOD*. The *ChaS* and *PAL* genes were expressed less in healthy plants than in esca-diseased ones, especially in August, but the phenolic compound values were far greater in the healthy grapevines. Possibly, the greater phenolic content in healthy leaves is due to the fact that both *PPO* expression and activity are at very low levels, which makes it impossible for these compounds to be degraded. There is also an effect on the redox homeostasis level, with a decrease in the AsA/GSH content. Increased *SOD* expression in esca-diseased leaves may act to alleviate the oxidative effects of O_2_.^−^, and could represent a SOD, PPO, and POD interaction as a defence system against the pathogen. Climatic conditions (high temperatures, water deficit) interact with esca-induced stress, altering the response. Therefore, the response to esca was more easily observable in leaves sampled in June, while, on the contrary, in leaves sampled in August, the observed effects would have been the result of both stressors.

## Figures and Tables

**Figure 1 antioxidants-11-01720-f001:**
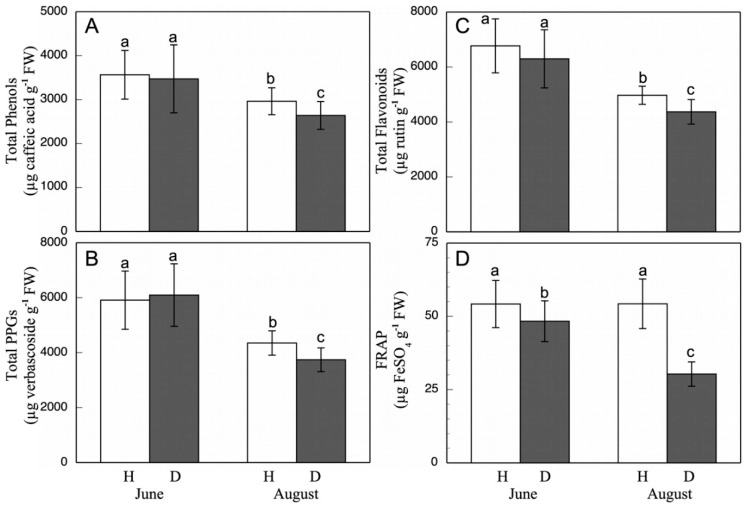
Total phenol (**A**), PPG (**B**), and flavonoid (**C**) contents, and FRAP (**D**) in grapevine healthy (H) and esca-diseased (D) leaves from June and August. The data are means ± SD from 10 independent experiments, each carried out in triplicate (different letters indicate significant differences at *p* < 0.05, Mann–Whitney U test).

**Figure 2 antioxidants-11-01720-f002:**
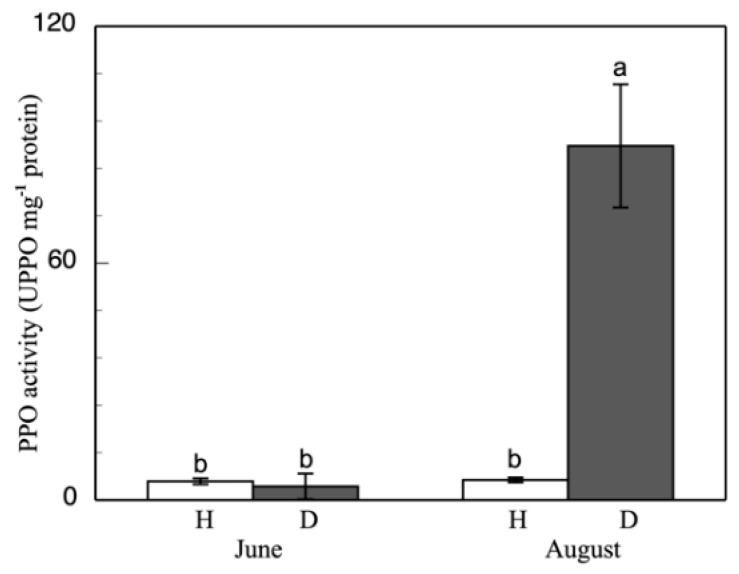
PPO activity in grapevine healthy (H) and esca-diseased (D) leaves from June and August. The data are means ± SD from 10 independent experiments, each carried out in triplicate (different letters indicate significant differences at *p* < 0.05, Mann–Whitney U test).

**Figure 3 antioxidants-11-01720-f003:**
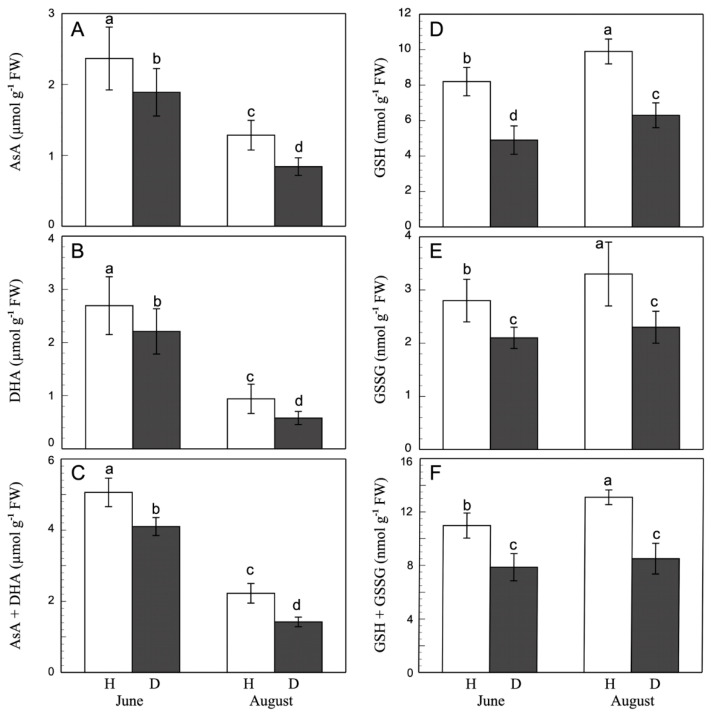
AsA (**A**), DHA (**B**), ascorbate pool (**C**), GSH (**D**), GSSG (**E**), and glutathione pool (**F**) in grapevine healthy (H) and esca-diseased (D) leaves from June and August. The data are means ± SD from 10 independent experiments, each carried out in triplicate (different letters indicate significant differences at *p* < 0.05, Mann–Whitney U test).

**Figure 4 antioxidants-11-01720-f004:**
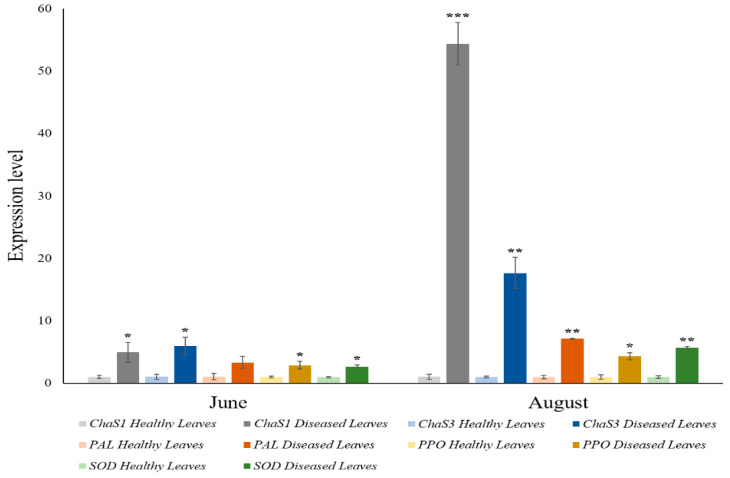
Transcript accumulation in grapevine healthy leaves vs. esca-diseased leaves, from June and August. Quantitative PCR in ChaS, ChaS3, PAL, PPO, and SOD of both types (healthy and esca-diseased plants) to quantify *ChaS*, *ChaS3*, *PAL*, *PPO*, and *SOD* transcripts. Significant differences by Student’s *t*-test between each transgenic line and Col-0 are marked (* *p* < 0.05; ** *p* < 0.05 ∩ *p* ≥ 0.001; *** *p* < 0.001).

**Figure 5 antioxidants-11-01720-f005:**
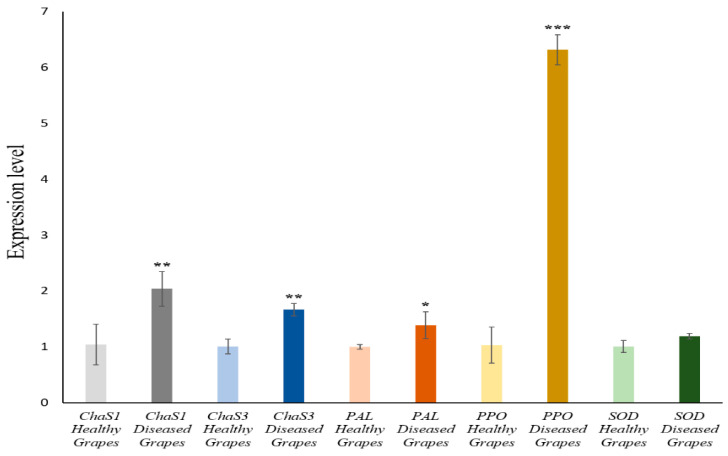
Transcript accumulation in grapevine healthy leaves vs. esca-diseased grapes, from October. Quantitative PCR in *ChaS*, *ChaS3*, *PAL*, *PPO*, and *SOD* of both types (healthy and esca-diseased plants) to quantify *ChaS*, *ChaS3*, *PAL*, *PPO* and *SOD* transcripts. Significant differences by Student’s t-test between each transgenic line and Col-0 are marked (* *p* < 0.05; ** *p* < 0.05 ∩ *p* ≥ 0.001; *** *p* < 0.001).

**Table 1 antioxidants-11-01720-t001:** Climatic conditions during the 2017 growth cycle. Monthly mean temperatures (T_max_, T_min_, and T_mean_), relative air humidity (RH_max_, RH_min_, and RH), radiation, net radiation, and rainfall.

	T_max_ (°C)	T_min_ (°C)	T_media_ (°C)	% RH_max_	% RH_min_	% RH	Radiation (MJ m^−2^ day^−1^)	Net Radiation (MJ m^−2^ day^−1^)	Rainfall (mm)
January	14.0	0.6	6.5	99.1	55.9	84.3	9.4	2.2	32.9
February	16.4	5.7	10.6	98.0	56.6	82.6	10.3	3.9	69.3
March	19.3	5.9	12.1	95.6	45.8	76.1	15.6	7.1	42.4
April	25.4	8.6	17.0	88.8	25.5	56.7	22.6	10.8	8.9
May	28.3	12.5	20.6	90.0	29.0	57.4	25.1	13.4	21.6
June	34.3	16.6	25.5	82.6	20.2	48.9	27.8	15.0	4.2
July	35.3	15.7	25.5	82.4	17.6	47.5	28.3	14.6	2.6
August	35.3	16.1	25.6	81.2	17.3	46.5	25.0	12.3	12.5
September	31.2	12.9	22.0	84.0	20.8	50.1	21.2	9.3	0
October	29.1	10.7	19.4	89.0	27.1	58.4	15.1	5.2	16.0
November	19.9	4.5	11.4	96.1	37.4	72.1	10.6	2.4	36.2
December	14.7	2.5	7.8	98.3	55.3	84.2	8.3	1.4	37.0
Mean annual	25.3	9.4	17.0	90.4	34.0	63.7	18.3	8.1	283.6 ^1^

^1^ Total annual rainfall.

**Table 2 antioxidants-11-01720-t002:** Sequences of the primers used for evaluation of expression level.

Gene	F/R	Sequence 5′-3′	Gene Information//Accession Number
*SOD*	F	CTGCGGGTTGGTGTTCTAAT	Superoxide dismutase, chloroplastic/cytosolic//
	R	TTCCCATATGGTGGTTCCAT	XM_002281814
*PAL*	F	ACAACAATGGACTGCCATCA	Phenylalanine ammonia lyase// XM_003633939
	R	GGAGGAGATTAAGCCCAAGG	
*PPO*	F	GGCTTTTCTTCCCTTTCCAC	Polyphenol oxidase, chloroplastic-like// XR_002029618
	R	ATTACAGTCGGAGGCAGGTG	
*Actin 1*	F	ACTGCTGAACGGGAAATTGT	*Actin* 2 (act2) mRNA Actin2-S1// AF369525
	R	AGTCCTCTTCCAGCCATCT	
*ChS1*	F	AGCCAGTGAAGCAGGTAGCC	Chalcone synthase 1// AB015872
	R	GTGATCCGGAAGTAGTAAT	
*ChS3*	F	GTTTCGGACCAGGGCTCACT	Chalcone synthase 3// AB066274
	R	GGCAAGTAAAGTGGAAACAG	
*VATP16*	F	CTTCTCCTGTATGGGAGCTG	V-type proton ATPase 16 kDa proteolipid subunit// XM_002269086
	R	CCATAACAACTGGTACAATCGAC	

**Table 3 antioxidants-11-01720-t003:** Effect of esca on the chlorophyll a and b and total chlorophyll contents, chlorophyll a/b ratio, total carotenoids, carotenoid/chlorophyll ratio, and photosynthetic efficiency (F_V_/F_M_) in leaves of *Vitis vinifera* cv. Tempranillo.

	Chl ^a^ (µg g^−1^ FW)	Chl ^b^ (µg g^−1^ FW)	Chl ^a+b^ (µg g^−1^ FW)	Chl ^a/b^	Carotenoids (µg g^−1^ FW)	Car/Chl	F_V_/F_M_
Healthy June	1686.9 ± 84.5 ^a^	844.8 ± 82.3 ^a^	2531.7 ± 166.9 ^a^	1.99 ± 0.09 ^b^	256.5 ± 17.9 ^a^	0.103 ± 0.010 ^b^	0.808 ± 0.050 ^a^
Esca-diseased June	865.9 ± 46.3 ^c^	401.9 ± 27.9 ^c^	1267.7 ± 74.3 ^c^	2.16 ± 0.03 ^a^	161.1 ± 3.5 ^b^	0.128 ± 0.005 ^a^	0.671 ± 0.062 ^b^
Healthy August	956.0 ± 60.1 ^b^	481.2 ± 24.5 ^b^	1437.2 ± 83.7 ^b^	1.99 ± 0.04 ^b^	119.6 ± 4.6 ^c^	0.084 ± 0.003 ^d^	0.781 ± 0.047 ^a^
Esca-diseased August	818.4 ± 38.6 ^c^	399.5 ± 19.8 ^c^	1217.9 ± 58.4 ^c^	2.05 ± 0.01 ^b^	115.2 ± 6.2 ^c^	0.094 ± 0.001 ^c^	0.475 ± 0.071 ^c^

Different letters indicate significant differences at *p* < 0.05, Mann–Whitney U test.

**Table 4 antioxidants-11-01720-t004:** Effect of esca on the membrane lipid peroxidation and proline content in leaves of *Vitis vinifera* cv. Tempranillo.

	Lipid Peroxidation (µmol MDA g^−1^ FW)	Proline Content (µg g^−1^ FW)
Healthy June	30.94 ± 0.68 ^b^	36.88 ± 1.95 ^c^
Esca-diseased June	28.80 ± 1.60 ^b^	71.82 ± 0.54 ^b^
Healthy August	41.54 ± 3.21 ^a^	118.15 ± 5.55 ^a^
Esca-diseased August	43.19 ± 2.34 ^a^	111.04 ± 67.71 ^a^

Different letters indicate significant differences at *p* < 0.05, Mann–Whitney U test.

**Table 5 antioxidants-11-01720-t005:** Effect of esca on the membrane lipid peroxidation, proline content, antioxidant capacity (FRAP) and PPO activity, total phenols, flavonoids, PPGs, and anthocyanin content in grapes of *Vitis vinifera* cv. Tempranillo.

	**Lipid Peroxidation** **(µmol MDA g^−1^ FW)**	**Proline Content** **(µg g^−1^ FW)**	**FRAP** **(µg g** **^−1^ FW)**	**PPO Activity** **(U mg** **^−1^ Protein)**
Healthy	75.53 ± 4.73 ^a^	591.73 ± 41.26 ^a^	53.04 ± 2.74 ^a^	26.36 ± 2.41 ^b^
Esca-diseased	41.75 ± 1.80 ^b^	579.50 ± 34.56 ^a^	27.17 ± 1.45 ^b^	232.06 ± 32.27 ^a^
	**Total Phenols** **(µg g^−1^ FW)**	**Total Flavonoids** **(µg g^−1^ FW)**	**Total PPGs** **(µg g^−1^ FW)**	**Total Anthocyanins** **(mg g^−1^ FW)**
Healthy	2359.98 ± 11.94 ^a^	3049.49 ± 217.32 ^a^	7730.54 ± 182.76 ^a^	451.02 ± 20.21 ^a^
Esca-diseased	1384.87 ± 83.45 ^b^	2343.33 ± 11.17 ^b^	5084.38 ± 179.72 ^b^	422.28 ± 36.10 ^a^

Different letters indicate significant differences at *p* < 0.05, Mann–Whitney U test.

**Table 6 antioxidants-11-01720-t006:** Effect of esca on the AsA, DHA, and ascorbate pool (AsA + DHA) contents, and the AsA/DHA ratio, the GSH, GSSG, and glutathione pool (GSH + GSSG) contents, and the GSH/GSSG ratio, in grapes of *Vitis vinifera* cv. Tempranillo.

	**AsA** **(µmol g^−1^ FW)**	**DHA** **(µmol g^−1^ FW)**	**AsA + DHA** **(µmol g^−1^FW)**	**AsA/DHA**
Healthy	1.48 ± 0.09 ^a^	0.58 ± 0.05 ^a^	2.06 ± 0.09 ^a^	2.83 ± 0.32 ^a^
Esca-diseased	0.49 ± 0.07 ^b^	0.42 ± 0.04 ^b^	0.91 ± 0.09 ^b^	1.22 ± 0.18 ^b^
	**GSH** **(nmol g^−1^ FW)**	**GSSG** **(µmol g^−1^ FW)**	**GSH + GSSG** **(µmol g^−1^ FW)**	**GSH/GSSG**
Healthy	12.88 ± 1.18 ^a^	2.22 ± 0.93 ^b^	15.04 ± 0.78 ^b^	11.91 ± 5.46 ^a^
Esca-diseased	12.18 ± 1.09 ^a^	5.33 ± 0.70 ^a^	17.51 ± 1.44 ^a^	2.59 ± 0.44 ^b^

Different letters indicate significant differences at *p* < 0.05, Mann–Whitney U test.

## Data Availability

The original contributions generated for this study are included in the article/Appendix A; further inquiries can be directed to the corresponding author.

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
