# Peer review of "Physiological and Molecular Responses of Vitis vinifera cv. Tempranillo Affected by Esca Disease"

_antioxidants, 2022, doi:10.3390/antiox11091720_

Round 1

Author Response

Response to Reviewer 1 Comments

  1. General consideration

The article has not been sent to, for example, the publication you indicate (Am J Enol Vitic, as we consider it more interesting to send it to a publication such as Antioxidants, which covers a wider range of subjects. It is not so much limited to the oenological and viticultural field as the magazine you cite. We believe that Antioxidants covers the aspects that we deal with in our work in a clearer way, without being limited to a specific aspect as it would be in Am J Enol Vitic, which is more related to the final quality of the wines obtained through cultural treatments.

  1. Abstract

According to the suggestion the indicated sentences have been rewritten, see pp 1, Abstract

  1. Keywords

According to suggestion, the new keywords have been

  1. Introduction

According your comments the sentence indicated has been modified

Apologizes, the first sentence indicating the meaning of GTD was wrong. The correct form has been included, clarifying the meaning of GTD

The indicated changes have been made: Has by have, Vides by vines, Vines by grapevines

Esca is the name of one of the fungal diseases that affect the grapevine, it is not an abbreviation

  1. Material and methods

Healthy leaves come from grapevines showing no esca symptoms in the last 3 years, and the leaves affected are those that come only from grapevines with visible symptoms, in tiger-like stage. The grapevines that have been used in this article were found in Agrarian Research Institute Finca “La Orden-Valdesequera” (CICYTEX). The grapevines were evaluated for more than 3 years and those responsible for this crop at CICYTEX carried out treatments against other pathogenic organisms (against Empoasca vitisTetranychus urticaePlasmopara viticola and Erysiphe necátor, for example) to ensure that only presented this type of pathology (esca), despite being an open field crop

Table 1 shows the data provided directly by the meteorological agency, we have not considered it necessary to eliminate the second decimal. In publications, such as the one you quote (Am J Enol Vitic) two decimals are included.

Apologizes, the red colour has eliminated

Thank you for helping us to improve our MS.

Kind regards,

The authors

Reviewer 2 Report

The research is focused on the physiological and molecular responses of Vitis vinifera cv. Tempranillo affected by Esca disease. The photosynthetic pigments, phenol, proline, ascorbate, and glutathione content, and the photosynthetic efficiency, antioxidant capacity, lipid peroxidation, and polyphenol oxidase activity have been evaluated.

In my opinion the paper is suitable for the publication in Antioxidants, with minor revision.

My concern is about:

1. General consideration:

Did the authors verify the actual presence of the pathogens involved in the symptoms? As they were working on a complex the PPO activity could vary depending on the presence of different pathogens.

Please report if the plants were protected from other foliar infections, such as downy mildew and powdery mildew, which could affect the PPO activity.

2. Abstract: In the last sentence of the abstract, the authors say that the expression of the genes, and the PPO activity, can be clearly used as markers of state in the development of the disease, but this may suggest the possibility to use these markers without the use of a healthy reference. Please eliminate “clearly”.

3. Statistical Analyses: the software used has not been indicated

4. Figure resolution and quality should be improved and standardized. Moreover, what the error bars are is not indicated.

5. Table 5 column “Proline content”: absence of statistic letters

6. The references must be carefully checked: some references lack some data or they lack italic characters in botanical names, there are capital letters instead of lowercase letters, etc.

7. The manuscript appears to have been drafted in a hurry, without a careful revision, and even a part written in red has been left.

The quality of the whole manuscript needs to be improved by correcting numerous writing errors, and having the final version reviewed by an English native speaker.

Specifically:

Row 17: please substitute “lowers” with decrease. Anyway, check the use of the verb “to lower” in the paper

Row 57: Phaeoacremonium and not Phaeoacreminium

Please check the syntax from row 95 to row 99

Row 115: please check in the paper all the sentences where the word “avoid” is reported (to avoid is a transitive verb but it is used as an intransitive verb)

Row 226: RNAse and not RNAsas

Row 228: “DNAsa” and “de” are Spanish words: please check them in the paper

Row 275: gene and not gen, check in the paper

Row 353: complex and not complez

Row 399: check “in addi-tion”Row from 429 to 432: check the sentence

Author Response

Response to Reviewer 2 Comments

  1. General consideration

The vine plants that have been used in this article were found in Agrarian Research Institute Finca “La Orden-Valdesequera” (CICYTEX). The vine plants were evaluated for more than 3 years and those responsible for this crop at CICYTEX carried out treatments against other pathogenic organisms (against Empoasca vitisTetranychus urticaePlasmopara viticola and Erysiphe necátor, for example) to ensure that the vines plants only presented this type of pathology (esca), despite being an open field crop.

  1. Abstract

According to suggestion, "clearly" has been deleted.

  1. Statistical analysis

According to suggestion the software used has been included, see pp 7, 2.10

  1. Resolution Figures

The figures have 300 ppp resolution (tiff archives)

  1. Table 5

Apologizes, in the table 5 the letters of signification (proline content) have been included

  1. References

The references have been reviewed and modified, according to your indications

  1. Minor revisions

A U.S. native has reviewed the manuscript and the grammatical errors they indicate have been fully corrected.

Has been replaced:

“lowers” by decrease

“RNAasa and DNAasa” by RNAse and DNAse

“Gen” by gene

“Complez” by complex

“Addi-tion” by addition

Phaeacreminium has been changed by Phaeoacreminium

“to avoid” has been corrected

“de” has been elimnate

All comments concerning the MS have been included.

Thank you for helping us to improve our MS.

Kind regards,

The authors

Round 2

Reviewer 1 Report

see attachment

Author Response

The whole manuscript has been corrected by an native English speaker.

  1. Title

According to suggestion, we have  modified the title we have add “esca disease”

  1. Abstract

Line 14, “with a complex” has deleted, and we modified the sentence

Line 15, we eraser “d” in evaluated

Line 16, we add the concrete location, “western Spain”

Line 20, we think that the word “affected” is more appropiate, because of we talk about symptomatology.  

Line 36, we changed the sentence

  1. Introduction

Line 42, apologies, it was a mistake.

Line 45, we did mention in abstract and line 82

Line xyz, we have decided to continue with GTD because, in our opinion, it is clear this abbreviature. We explained that abbreviature in line 36.

Lines 59-69, After an exhaustive revision of the literature we didn´t found any reference reporting a clear relationship between iron concentration and esca. Particulary, Calzarano et al. (2017)[ Phytopathologia Mediterranea (2017) 56, 3, 494−501 DOI: 10.14601/Phytopathol_Mediterr-22055], demonstrated the inexistence of a relationship between iron and chlorosis caused by esca. However, an older paper (Di Marco et al, 2001; Phytopathol_Mediterr  2001, 40, supplement S449-S452; DOI: 10.14601/Phytopathol_Mediterr-1624) meassured a higher content of iron in symptomatic leaves when compared with the asymptomatic ones. Recently, Ouadi et al. (2019) [PLoS ONE 14(9): e0222586. https://doi.org/10.1371/ journal.pone.0222586] report about a decrease of nitrogen content in leaves from plants with esca disease.

Lines 95 and 98, we made the proposed changes

Table 1, We followed the recommendations made by the referee

Thank you for helping us to improve our MS.

Kind regards,

The authors

Round 3

Reviewer 1 Report

The authors have for the first time improved the manuscript- a few formal language issues remain before the manuscript can be accepted such as

1)     Title, insert “fungal” after esca to separate from virus and bacterial infections

Abstract

2)     line 15, insert “grapevine“ before cv. Tempranillo

3)     line 16, insert “in the leaves” after pigments (if correct=?)

4)     lines 17, 20 and 2xdecrease => decreased (add -d), past tense and

in line 17 increases => increased

5)   use p a s t tense throughout the abstract

Intro:

6)     line 40, remove “is” (Grammar)

7)     Line 45, was=>is

The manuscript remains difficult to understand unless you work on this fungus

Author Response

Dear Reviewer,

Thank you very much for your suggestions and advices. All your comments have been very helpful to us.

  1. We do not consider necessary to add "fungal" in the title, because from our point of view it would becames redundant. On the other hand, the fungal origin is indicated both in the abstract and in the introduction. Similar articles do not add the term fungal in their title, since they are published in very different journals (Moret F, Lemaître-Guillier C, Grosjean C, Clément G, Coelho C, Negrel J, Jacquens L, Morvan G, Mouille G, Trouvelot S, Fontaine F and Adrian M (2019) Clone-Dependent Expression of Esca Disease Revealed by Leaf Metabolite Analysis. Front. Plant Sci. 9:1960. doi: 10.3389/fpls.2018.01960; Christen, D.; Schönmann, S.; Jermini, M.; Strasser, R.J.; Défago, G. Characterization and early detection of grapevine (Vitis vinifera) stress responses to esca disease by in situ chlorophyll fluorescence and comparison with drought stress. Environ. Exp. Bot. 2007, 60, 504–514, doi:10.1016/j.envexpbot.2007.02.003; Calzarano, F.; Pagnani, G.,; Pisante, M.; Bellocci, M.; Cillo, G.; Metruccio, E.G.; Di Marco, S. Factors involved on tiger-stripe foliar symptom expression of esca of grapevine. Plants 2021, 10, 1041, https://doi.org/10.3390/ plants10061041; Foglia, R.; Landi, L.; Romanazzi, G. Analyses of Xylem Vessel Size on Grapevine Cultivars and Relationship with Incidence of Esca Disease, a Threat to Grape Quality.  Sci.2022, 12, 1177. https://doi.org/10.3390/app12031177).

2-7) All grammatical changes that you propose are done. 

Kind regards,

The authors